# Structural basis of anticancer drug recognition and amino acid transport by LAT1

Yongchan Lee [1,2] ✉, Chunhuan Jin[3], Ryuichi Ohgaki [3,4], Minhui Xu[3], Satoshi Ogasawara [5], Rangana Warshamanage[6], Keitaro Yamashita [7], Garib Murshudov[7], Osamu Nureki [8], Takeshi Murata [5] & Yoshikatsu Kanai [3,4,9] ✉

LAT1 (SLC7A5) transports large neutral amino acids and plays pivotal roles in cancer proliferation, immune response and drug delivery. Despite recent advances in structural understanding of LAT1, how it discriminates substrates and inhibitors including the clinically relevant drugs remains elusive. Here we report six structures of LAT1 across three conformations with bound ligands, elucidating its substrate transport and inhibitory mechanisms. JPH203 (also known as nanvuranlat or KYT-0353), an anticancer drug in clinical trials, traps LAT1 in an outward-facing state with a U-shaped conformer, with its amino-phenylbenzoxazol moiety pushing against transmembrane helix 3 (TM3) and bending TM10. Physiological substrates like L-Phe lack such effects, whereas melphalan poses steric hindrance, explaining its inhibitory activity. The "classical" system L inhibitor BCH induces an occluded state critical for transport, confirming its substrate-like behavior. These findings provide a structural basis for substrate recognition and inhibition of LAT1, guiding future drug design.

Amino acid transporters are essential for maintaining cellular amino acid homeostasis and are integral to various metabolic and signaling pathways[1]. Among them, the L-type amino acid transporter 1 (LAT1; SLC7A5) is notable for its broad substrate specificity, transporting not only essential neutral amino acids such as L-Leu and L-Phe, but also the basic amino acid L-His, non-essential neutral amino acids L-Cys and L-Tyr, bulkier derivatives such as thyroid hormones (e.g., $T_4$ and $T_3$), and amino acid mustards such as melphalan (Fig. 1a)[2–4]. This broad substrate spectrum underscores the critical role of LAT1 in transporting essential nutrients, biomolecules and therapeutic agents across the cell membrane[5]. LAT1 belongs to the classically characterized System L, the major amino acid transport system that prefers neutral amino acids[6]. Among the four molecular entities that constitute System L (LAT1–4), LAT1 and LAT2 belong to the SLC7 family and form heterodimeric complexes with the single membrane-spanning protein CD98hc (4F2hc; SLC3A2)[7]. In contrast, LAT3 and LAT4 belong to the SLC43 family and do not form heterodimers with CD98hc[8].

[1]Department of Structural Biology, Max Planck Institute of Biophysics, 60438 Frankfurt, Germany. [2]Graduate School of Medical Life Science, Yokohama City University, Yokohama, Kanagawa 230-0045, Japan. [3]Department of Bio-system Pharmacology, Graduate School of Medicine, Osaka University, Osaka 565-0871, Japan. [4]Integrated Frontier Research for Medical Science Division, Institute for Open and Transdisciplinary Research Initiatives (OTRI), Osaka University, Osaka 565-0871, Japan. [5]Graduate School of Science, Chiba University, Chiba 263-8522, Japan. [6]Scientific Computing Department, UKRI Science and Technology Facilities Council, Rutherford Appleton Laboratory, Harwell Campus, Didcot OX11 0FA, UK. [7]Structural Studies Division, MRC Laboratory of Molecular Biology, Cambridge CB2 0QH, UK. [8]Department of Biological Sciences, Graduate School of Science, The University of Tokyo, Tokyo 113-0033, Japan. [9]Premium Research Institute for Human Metaverse Medicine (WPI-PRIMe), Osaka University, Osaka 565-0871, Japan. ✉e-mail: yongchan.lee@biophys.mpg.de; ykanai@pharma1.med.osaka-u.ac.jp

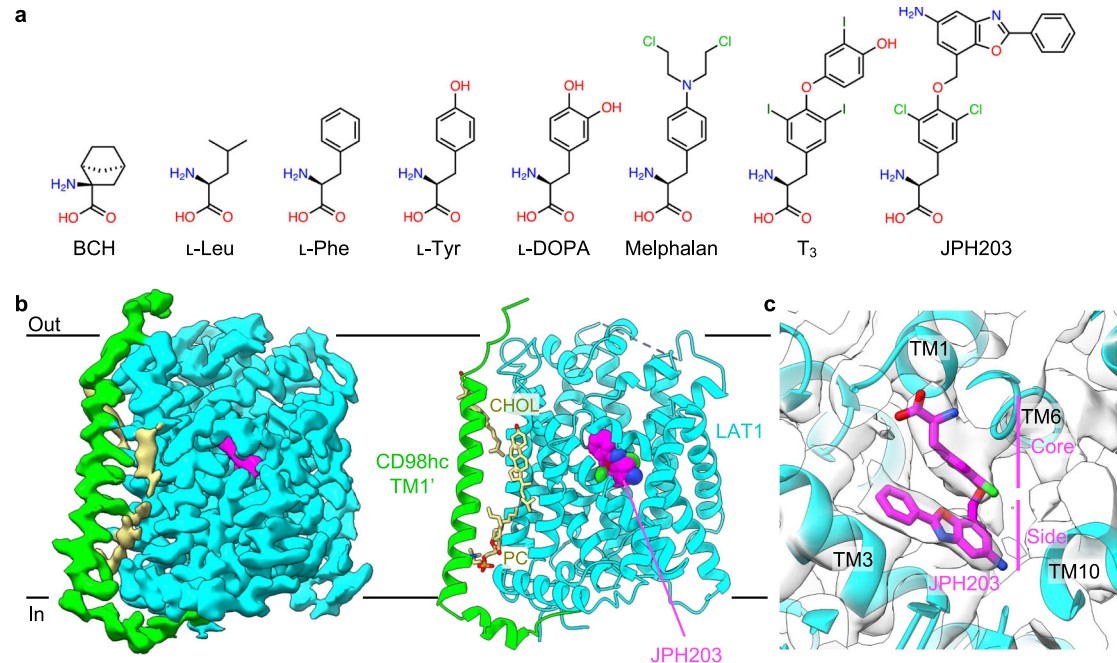

**Fig. 1 | Structure of LAT1 bound to JPH203. a** Chemical structures of selected LAT1 substrates and inhibitors. **b** Cryo-EM map and the overall structure of LAT1 bound to JPH203. The map of the transmembrane domain after local refinement is shown. Nanodisc densities are not displayed for clarity. **c** Close-up view of the JPH203-binding site overlayed with the map.

Since LAT1 is upregulated in various types of cancer, its inhibitors are recognized as potent anti-tumor agents[5]. In addition, LAT1 radio-tracers are used in the positron emission tomography (PET) and the single photon emission computed tomography (SPECT) for cancer imaging[9,10], and the boronated substrates are used in boron neutron capture therapy (BNCT) for cancer treatment[9,10]. LAT1 is also responsible for the transport of essential amino acids[11] and the amino acid-derived drugs and pro-drugs across the blood-brain barrier (BBB), such as gabapentin and L-DOPA (Fig. 1a), prescribed for epilepsy[12] and Parkinson's disease[13], respectively, making it essential for the delivery of nutrients and drugs to the brain.

JPH203 (Fig. 1a) is a high-affinity inhibitor of LAT1 with a sub-micromolar $IC_{50}$ and no detectable inhibition on LAT2[14]. With such outstanding selectivity and potency, JPH203 has been proved effective against different types of cancers[15,16] and successfully completed Phase I and II clinical trials as a first-in-class drug against biliary tract cancer[17,18]. Although its structural design is inspired by $T_3$, JPH203 has a bulkier hydrophobic side chain, which may be responsible for its high selectivity and affinity (Fig. 1a). 2-Aminobicyclo-(2,2,1)-heptane-2-car-boxylic acid, also known as BCH, is a "classical" system L inhibitor with broad specificity towards both LAT1 and LAT2, as well as other system L[1]. BCH has a bicyclic norbornane moiety and is significantly smaller than other Tyr-based inhibitors (Fig. 1a).

A recent structural study of LAT1 showed that JPH203 did not bind to their purified LAT1[19], presumably due to the presence of detergent, which hindered the structural understanding of JPH203. Additionally, the same study reported a LAT1 structure with a BCH ligand modeled; however, the ligand density was very weak and indistinguishable from the apo map, and caution is required when interpreting this model (Supplementary Fig. 1, difference maps). Therefore, the inhibition mechanism by BCH remains to be elucidated. More recently, JX compounds, which are bicyclic meta-Tyr derivatives, have been reported as high-affinity inhibitors. The associated cryo-EM structures revealed how these compounds bind to the outward-occluded conformations of LAT1[20]. This was the first structural demonstration of system L inhibition. However, the different core structures of JX inhibitors compared to JPH203 and other amino acid-like compounds have hindered the understanding how these well-known system L inhibitors bind to and inhibit LAT1.

Here, we employ the lipid nanodisc system and electron cryo-microscopy (cryo-EM) to study the structure of LAT1. The structural analyses accompanied by the functional assays of site-directed mutants illuminate how LAT1 dynamically interacts with transportable and non-transportable compounds, including JPH203.

## Results

### LAT1 in lipid nanodiscs

We hypothesized that lipid environment is the key to investigating system L transport and inhibition. We purified LAT1–CD98hc and reconstituted it into nanodiscs (Supplementary Fig. 2a), with a phospholipid mixture supplemented with cholesterol, which has been shown to be important for transport activity[21,22]. To add fiducial markers for single-particle analysis, we generated mouse monoclonal antibodies and screened structure-specific binders by ELISA, FACS and negative-stain electron microscopy. The Fab fragment from clone 170 (Fab170) was found to bind to CD98hc (Supplementary Fig. 2b, c), similar to a previously characterized antibody MEM-108[23], and was used for subsequent studies. These technical improvements and sample optimization led to the structure determination of LAT1–CD98hc bound to JPH203 by cryo-EM at nominal resolution of 3.9 Å. Focused refinement on the transmembrane domain (TMD) yielded a map with better density for the ligand, with local resolutions extending to 3.6 Å (Supplementary Fig. 2d–q and Supplementary Tables 1 and 2).

### JPH203 binds to the outward-facing pocket of LAT1

The structure of LAT1 bound to JPH203 shows an outward-facing conformation with the extracellular halves of TM1 and TM6 (named TM1b and TM6a) widely open (Fig. 1b). The cryo-EM map shows a well-resolved density for JPH203, which adopts a U-shaped conformer and is stuck between the hash and the bundle domains (Fig. 1c). Its α-carboxy and α-amino groups are recognized by the unwound regions

of TM1 and TM6, respectively, agreeing well with the proposed binding modes for amino acid substrates (Fig. 1c). Unexpectedly, the 5-amino-2-phenylbenzoxazol side group (Fig. 1c) is not accommodated in the previously proposed "distal pocket" surrounded by TM6 and TM10[23], but instead faces TM3 in an opposite direction and is partially exposed to the extracellular solvent (Fig. 2a). The three aromatic rings align nearly parallel to each other and face a flat hydrophobic patch formed by Ile140, Ser144, Ile147, Val148 and Ile397 on TM3 and TM10 (Fig. 2a). The core dichloro-Tyr moiety of JPH203 is sandwiched by two aromatic residues: Phe252 forms a T-shaped π–π interaction with the terminal phenyl moiety of JPH203 on the extracellular side and Tyr259 forms halogen bonding interaction with a chloride atom of JPH203 (Fig. 2b, c). Furthermore, JPH203 introduces a kink on TM10 with its nitrogen atom wedging into the helix (Fig. 2a). Together, these interactions widen the substrate binding pocket of LAT1 and fix it in the wide, outward-open conformation.

JPH203 is known for its outstanding selectivity for LAT1 over LAT2[14]. To analyze how selectivity is conferred, we compared amino acid sequences of LAT1 and LAT2, focusing on both the JPH203-interacting residues and the surrounding regions, and found key differences (Fig. 2c). For instance, Ser144 is substituted to Asn in LAT2, which may hinder the accommodation of the side group, and Phe400 to Val, which may alter hydrophobic interactions within and around the binding pocket. To test how these residues influence inhibitor sensitivity, we generated single or double variants of LAT1 and evaluated their inhibition by JPH203 using radioactive L-Leu uptake assays in *Xenopus* oocytes

(Supplementary Fig. 4). Among several variants tested, the double variant F400V/S144N, which retained ~20% of the maximal L-Leu transport activity of the wild-type, altered the sensitivity to JPH203, with almost negligible inhibition even at 1 μM, suggesting that one of these residues may be important for JPH203 selectivity (Supplementary Fig. 4a, b). Although a single variant S144N abolished L-Leu transport, and thus its own effect on inhibition could not be evaluated independently (Supplementary Fig. 4a), the variant F400V retained its activity and was sensitive to JPH203 (Supplementary Fig. 4a, b), leaving Ser144 as a key residue for JPH203 selectivity. We reason that the two mutations of opposite effects (smaller to larger and larger to smaller) compensated for each other to restore the binding pocket for amino acid substrates (as seen in the LAT2 structures[24,25]), whereas this compensation was not sufficient for JPH203 due to stricter recognition. To further investigate the structural features involved in JPH203 recognition, we examined the effects of other amino acid substitutions (Supplementary Fig. 4c). Surprisingly, an F252W variant showed more potent inhibition by JPH203 than the wild-type, with the $IC_{50}$ value decreasing from 542 nM to 36 nM (Fig. 2d, e). This may be explained by enhanced aromatic interaction between the introduced tryptophan and the terminal phenyl moiety of JPH203, consistent with the observation that adding a methoxy group to the terminal ring of JPH203 improves its affinity[26], presumably by enhancing the aromaticity.

We next compared the structure of LAT1 bound to JPH203 with those bound to JX-075, JX-078 and JX-119[20]. These compounds are bound to the same site in the pocket, but their detailed binding poses

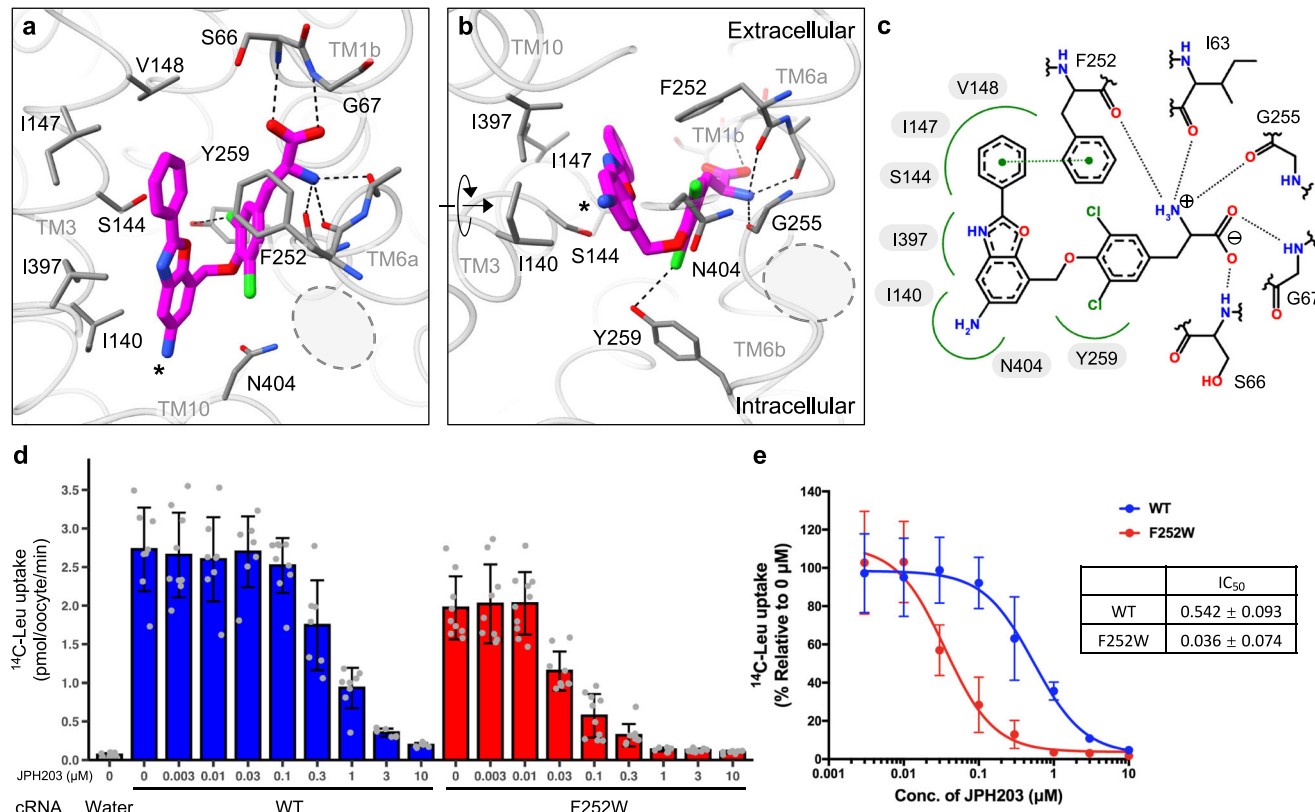

**Fig. 2 | Structural basis of LAT1 inhibition by JPH203. a, b** Close-up views of the JPH203-binding site. Interacting residues are shown as stick models. Hydrogen bonds are depicted as dotted lines. The 5-amino group of JPH203 is marked by an asterisk. The previously predicted "distal pocket" is marked by gray dotted circles. **c** Schematic diagram of the interactions between JPH203 and LAT1. Hydrogen bonds are depicted as black dashed lines. Hydrophobic contacts are depicted by green splines. A π–π interaction is depicted as a green dashed line connecting two green dots. **d** Inhibition of LAT1 by JPH203 at different concentrations. Uptake of L-[¹⁴C]Leu into *Xenopus* oocytes expressing co-expressing CD98hc and wild-type or

F252W LAT1 was measured in the presence of JPH203 at indicated concentrations. As a negative control, water was injected instead of cRNAs. Data are mean ± SD and each data point represents a single oocyte (n = 7 for 0.03, 0.3, 1 and 10 μM for WT; n = 8 for 0, 0.01 and 3 μM for WT and 3 and 10 μM for F252W; n = 9 for 0.003 and 0.1 for WT and 0, 0.003, 0.03, 0.1 and 1 for F252W; and n = 10 for 0.01 and 0.3 for F252W). **e** Concentration-dependent inhibition curves by JPH203 for wild-type or F252W LAT1, calculated using the same data as shown in panel (**d**). Source data are available with this paper as Source Data file.

and structural effects on LAT1 are different (Supplementary Fig. 3a–c). The core 2-amino-1,2,3,4-tetrahydro-2-naphthoic acid moiety of JX inhibitors is positioned deeper in the pocket and closer to TM3, by about 2 Å as compared to the equivalent moieties of JPH203 (Supplementary Fig. 3a). In addition, the gating bundle (TM1b and TM6a) adopts a more closed conformation in the JX series compared to JPH203 (Supplementary Fig. 3a). Among the three JX structures reported, only JX-119 adopts a U-shaped conformation similar to JPH203, although the blurred ligand density suggests that its terminal moiety is flexible and lacks the critical T-shaped π-π stacking with Phe252 observed in JPH203 (Supplementary Fig. 3b, c). Interestingly, the structure of LAT1 bound to diiodo-Tyr[20] aligns well with the core Tyr moiety of JPH203 (Supplementary Fig. 3c). Both compounds share two halogen atoms at 3'- and 5'-positions, which appear to contribute to the large opening angle of TM1b and TM6a (Supplementary Fig. 3c). However, due to the absence of a bulky side group, in the diiodo-Tyr-bound structure, TM10 adopts a straight form[20], rendering the substrate-binding site not as widely open as in the JPH203-bound

structure (Fig. 2a). Taken together, both the bi-halogenated Tyr core and the bulky side group of JPH203 contribute to the maximal opening of TM1b, TM6a, TM3 and TM10 to ensure the wide outward-open conformation of LAT1 observed here.

## Phe and Melphalan show different degrees of interactions within the pocket

Given the successful structure determination of LAT1 bound to JPH203 in nanodiscs, we next determined the structures of LAT1 bound to a physiological substrate L-Phe and a slow substrate melphalan (Fig. 3a–d and Supplementary Fig. 2 f, g). We also serendipitously obtained an apo outward-open structure from the sample incubated with T₃ for a short time, which served as a reference for ligand density validation (Supplementary Fig. 2e, h, 5; also see "Methods"). All three structures adopt a similar outward-facing conformation, in which TM1b and TM6a are widely open with slightly different angles (Fig. 3a–e). In the apo outward-open structure, Phe252 is flipped "up", exposing the substrate-binding pocket towards the extracellular solvent (Fig. 3e, h).

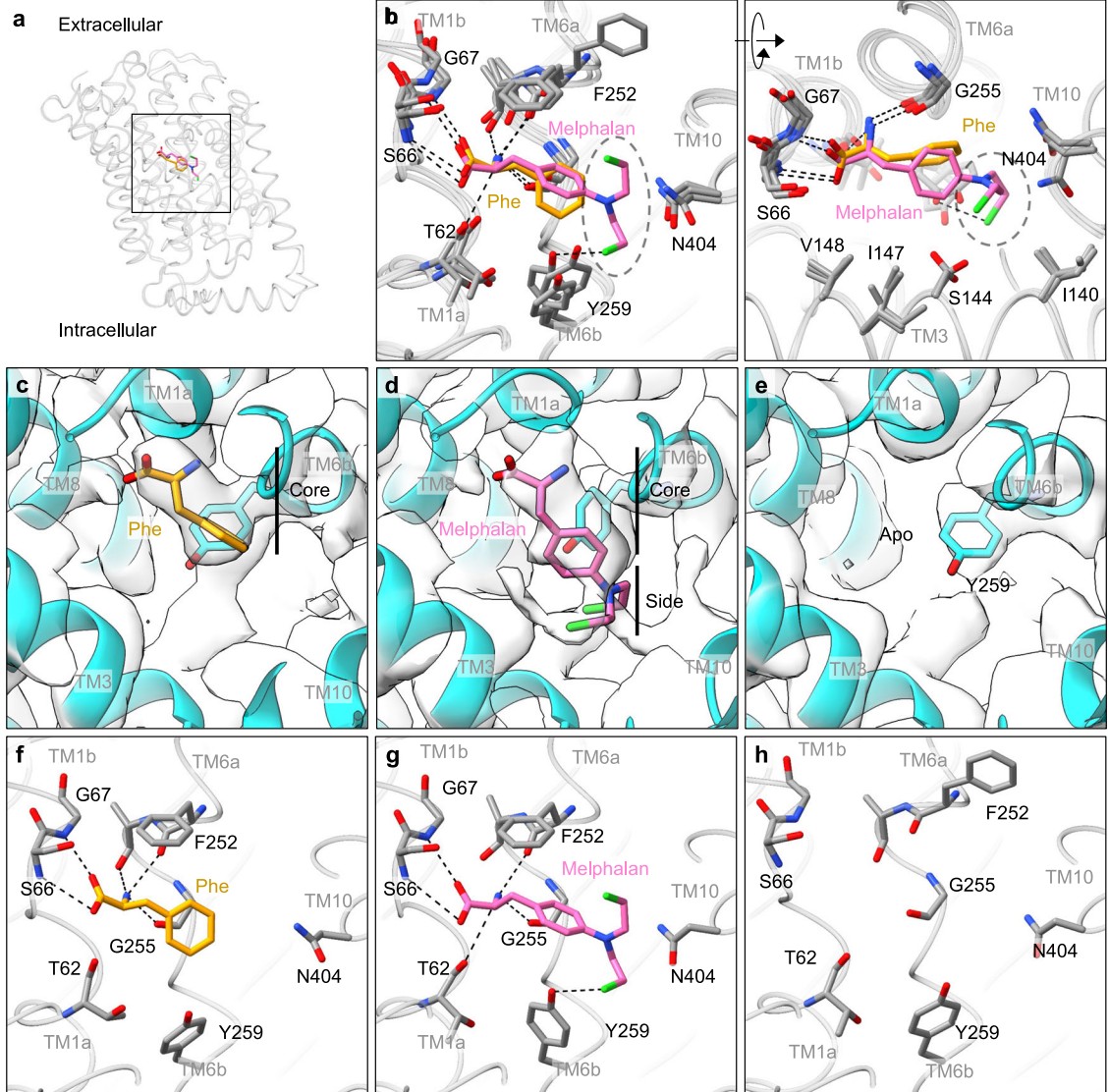

**Fig. 3 | Cryo-EM structures of LAT1 bound to Phe and melphalan or in apo in the outward-facing conformation. a** Overlay of the Phe- and melphalan-bound LAT1, with the ligands displayed as stick models. The square indicates the region zoomed up in panels (**b**–**h**). **b** Superposition of the Phe-bound, melphalan-bound and apo LAT1 structures, viewed from two different angles. The dotted circles highlight the bis-(2-chloroethyl)amino group of melphalan, accommodated in a hydrophobic space created by TM3, TM6a and TM10. **c, f** Structure of Phe-bound LAT1 in the outward-open conformation. **d, g** Structure of melphalan-bound LAT1 in the outward-open conformation. **e, h** Structure of apo LAT1 in the outward-open conformation.

In the L-Phe- and melphalan-bound structures (Fig. 3f, g), the binding poses of the ligands agree well with previous predictions[27] and that of JPH203, where the substrate carboxy and amino groups are recognized by the exposed main chain atoms of TM1 and TM6, respectively. The phenyl ring of L-Phe faces Gly255 and forms van der Waals interactions with its Cβ atom (Fig. 3f), consistent with our previous finding that Gly255 is important for recognition of larger amino acids[23]. Melphalan shows similar interactions in the core, but the additional bis-(2-chloroethyl)amino side group is placed in the space surrounded by TM3, TM6a and TM10 to form further interactions (Fig. 3g). Although the moderate local resolution of the map (~3.9 Å) obscures precise positioning of individual atoms (Fig. 3d and Supplementary Fig. 5), each of the two chloroethyl moieties appears to point upwards and downwards, with terminal chloride atoms within halogen-bonding distances to Tyr259 and Asn404 (Fig. 3g). These additional interactions would add steric hindrance to TM3, TM6b and TM10 and restrict the conformational change, which could explain its slow transport rate across the blood-brain barrier[28] and its inhibitory action on amino acid transport[3].

Among alkylating agents, melphalan and other amino acid-mustards have been extensively studied for their interaction with LAT1. These agents are known to be transported at different rates and sometimes act as potent inhibitors, depending on their core structure, the position of the mustard moiety and other substitutions[29]. For example, phenylglycine mustard (PGA) is transported at a higher rate than melphalan[29], whereas *meta*-substituted phenylalanine mustard derivatives act as potent inhibitors[30,31]. Our structure suggests that the shorter core structure of PGA would create a weaker steric hindrance around TM10, allowing for faster transport, whereas bulkier substitutions could create significant steric hindrance, enhancing inhibitory properties, especially at the *meta*-position.

## Non-selective system L-inhibitor BCH induces an occluded state

We next determined the structure of LAT1 bound to the "classical" system L inhibitor BCH at 3.7 Å resolution (Fig. 4a, b). Intriguingly, unlike all the other inhibitor-bound structures, the BCH-bound LAT1 is captured in an occluded state, a previously unseen conformation where the ligand is fully sealed from both sides of the membrane (Fig. 4c). We note that this conformation differs from the previously reported "BCH-bound", inward-open structure[19], in which the modeled BCH shows poor density, making it indistinguishable from the structure without any substrate (Supplementary Fig. 1). BCH is bound in the canonical substrate-binding pocket, with its carboxy and amino moieties recognized by the exposed main-chain atoms of TM1 and TM6 (Fig. 4b). In comparison to the JPH203-bound structure (Fig. 4d), the pocket of BCH-bound LAT1 is significantly narrowed (Fig. 4e). TM1b and TM6a move closer to TM3, allowing the hydrophobic norbornane moiety of BCH to contact Ser144, Ile147 and Val148. In addition, Phe252 sits on top of BCH, completely occluding it from the extracellular solvent (Fig. 4c, e). Furthermore, TM10 is twisted by about 70° to form a straight helix (Fig. 4f), bringing Phe400 into contact with the side face of the norbornane moiety. These collective structural changes collapse the space occupied by the 5-amino-2-phenylbenzoxazol moiety of JPH203, making the pocket just large enough to accommodate BCH (Fig. 4c).

## JPH203 blocks transport while BCH induces antiport

In the "alternating-access" scheme of membrane transporters[32], an occluded state represents the key intermediate that connects the outward- and inward-open states to enable substrate translocation[33]. In antiporters, the formation of an occluded state must be coupled to substrate binding, which lowers the energy barrier for structural isomerization[32]. Our observation that BCH induces such an occluded state suggests that BCH may trigger state transitions and advance the critical backward isomerization step in the antiport cycle of LAT1.

Indeed, a previous study in HEK293 cells has shown that BCH can induce amino acid efflux from the *trans* side, acting as a counter-substrate for antiport[27]. To confirm this, we performed similar transport assays using *Xenopus* oocytes expressing LAT1 and CD98hc. In *cis* inhibition assays, in which the inhibitor is present in the external solution, BCH inhibited the uptake of the radioactive L-Leu into oocytes (Fig. 4g). This effect was similar to that of JPH203, although BCH required much higher concentrations to achieve a similar level of inhibition (Fig. 4g). We next performed counterflow assays by first pre-loading radioactive L-Leu into oocytes and then monitoring the efflux of radioactivity upon application of compounds of interest in the external solution (Fig. 4h). The results showed that extracellular BCH indeed induced L-Leu efflux from the cell to the external solution (Fig. 4h). The total efflux was comparable to that stimulated by the physiological substrate L-Leu (Fig. 4h). In contrast, JPH203 did not induce efflux (Fig. 4h), consistent with its blocking action from the extracellular side (Fig. 1b). These observations confirm that BCH is a transportable substrate of LAT1, which acts as a competitive inhibitor when present with substrates on the same side of the membrane. This action is in stark contrast to JPH203, which, as shown above, is a bona fide blocker acting on the extracellular side of LAT1.

## Structural rearrangements in the transport mechanism

During image processing, we found that cryo-EM data of LAT1 incubated with some of the ligands contained subpopulation of particles representing the inward-open structures (Supplementary Fig. 2h, i, j). In all of these cases, we observed only the empty substrate-binding site, suggesting that the inward-open state is a low-affinity state that favors ligand release. We also collected cryo-EM data in the absence of any ligands and observed only the apo inward-facing particles (Supplementary Fig. 2k, n), suggesting that this is the ground state preferred by LAT1 without a substrate, consistent with the previous apo inward-open structures observed in detergent[19,23]. By combining all these particles, we obtained the apo inward-open structure at 3.6 Å resolution (Supplementary Fig. 2l, p), completing all major conformations of LAT1 in the lipid nanodiscs and allowing detailed investigation of structural rearrangements during substrate transport (Fig. 5).

Major structural changes occur in the gating bundle, in agreement with other well-studied LeuT-fold transporters (Fig. 5a)[32]. In the outward-open to occluded state transition, TM6a and TM1b incline towards the hash domain by about 18° (Fig. 5a), and Phe252 sits atop the substrate, sealing the extracellular vestibule (Fig. 4e). TM10 undergoes helix rearrangements that include the rotation of about 160°, bringing Phe400 into direct hydrophobic contact with the substrate (Fig. 5b). Accompanied by these, EL4 moves closer to the CD98hc ectodomain. In the occluded to inward-facing state transition, TM6a and TM1b undergo further inward rotation towards the hash domain, accompanied by the movements of EL4, which now meets EL2 and CD98hc (Fig. 5a). TM10 undergoes a further shift of about 3 Å, to tighten the extracellular gate. Upon adopting the inward-open state, TM1a and TM6b swing open, creating an intracellular vestibule that would allow the release of substrates inside the cell (Fig. 5a, c). The binding of a counter-substrate from inside the cell would trigger the reverse process to complete an antiporter turnover.

We found that the *N*-terminal residues 44–50, which are disordered in the inward-facing conformations[19,23], form a defined structure in the outward-open and occluded conformations (Fig. 5e). This region forms multiple hydrophilic and hydrophobic interactions at the interface of the hash and bundle domains to strengthen the cytoplasmic gate, suggesting that ordering and disordering of this region are involved in the conformational changes of LAT1 for substrate transport. To test this, we generated an *N*-terminal truncation variant of LAT1, designated Δ1–50. This variant retained cell surface expression in oocytes (Supplementary Fig. 6b, c), but showed no transport activities (Fig. 5f), demonstrating the importance of these residues for

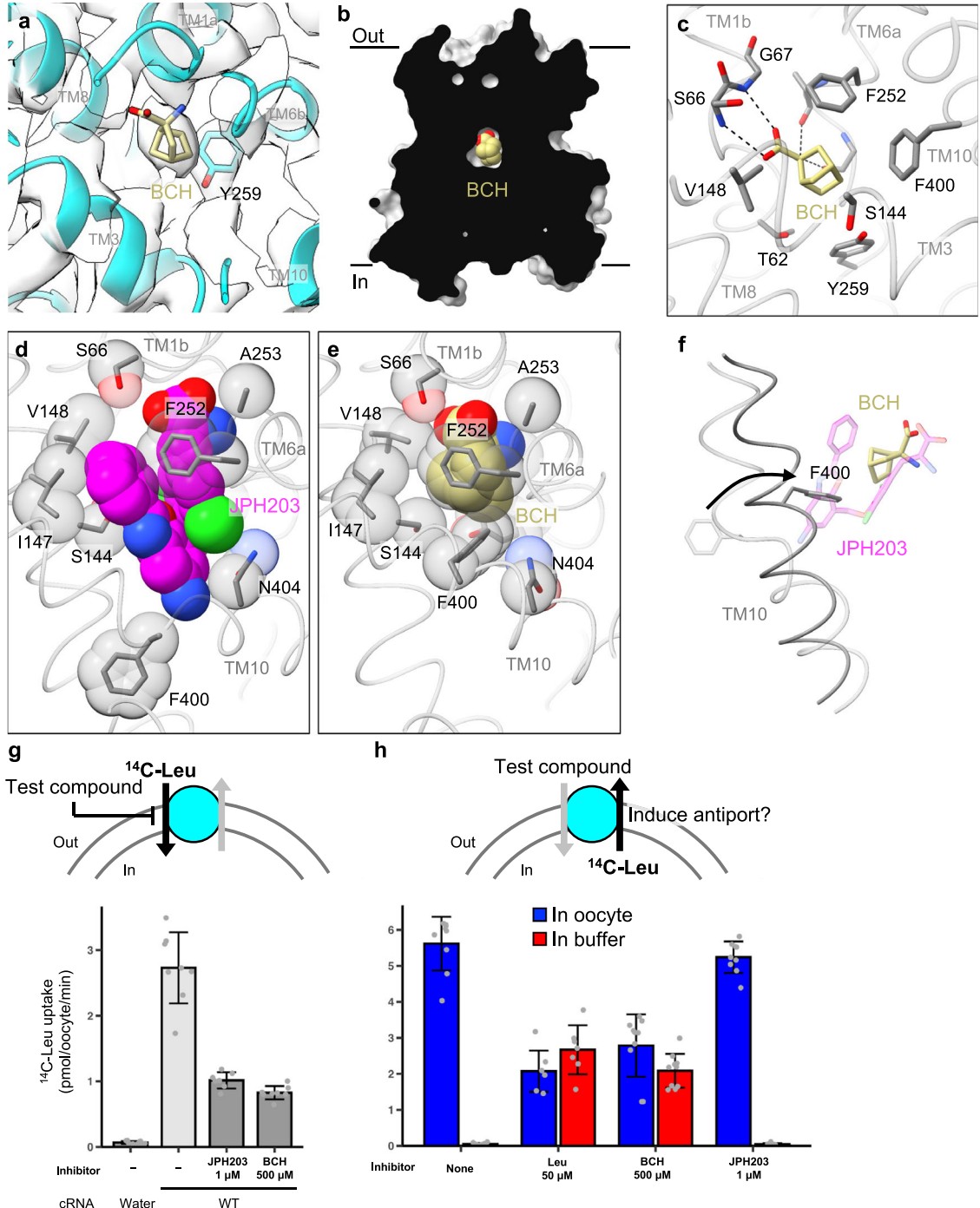

**Fig. 4 | LAT1 bound to BCH in the occluded conformation. a** Close-up view of BCH modeled into the cryo-EM map. **b** Cut-away surface representation of LAT1, showing that BCH is occluded from both sides of the membrane. **c** Interaction of BCH with the surrounding residues. Dotted lines depict hydrogen bonds. **d**, **e** Comparison of JPH203- and BCH-binding sites of LAT1. The ligands and important residues are shown as sticks and spheres. **f** Rotation of Phe400 from the JPH203-bound (transparent) to BCH-bound (opaque) structures. **g** *cis* inhibition assay. Uptake of L-[¹⁴C]Leu into *Xenopus* oocytes expressing wild-type LAT1 and CD98hc was measured in the presence or absence of indicated compounds in the external solution. As a control, water was injected instead of cRNAs. Data are mean ± SD and each data point represents a single oocyte (*n* = 7 for 1 µM JPH203; *n* = 8 for 0 µM JPH203 and 500 µM BCH; and *n* = 10 for Water). **h** Counterflow assay to evaluate substrate-induced antiport. Efflux of pre-loaded L-[¹⁴C]Leu from *Xenopus* oocytes was measured in the presence or absence of indicated compounds in the external buffer solution. Data are mean ± SD and each data point represents a single oocyte (*n* = 7 for Leu; *n* = 9 for buffer and JPH203; and *n* = 10 for BCH). Source data are available with this paper as Source Data file.

the catalytic mechanism. We also tested several other variants of this region but observed varying degrees of cell-surface localizations, which hampered identification of critical residues for substrate translocation. The deleted *N*-terminal 50 residues include three lysine residues (Lys19, Lys25 and Lys30) that have been shown to be

important for LAT1 downregulation through ubiquitination[34]. Our results suggest another important role of this cytoplasmic region in the transport mechanism.

Notably, we observed the numerous lipid densities surrounding the TMD of LAT1–CD98hc (Supplementary Fig. 7). Of these, one

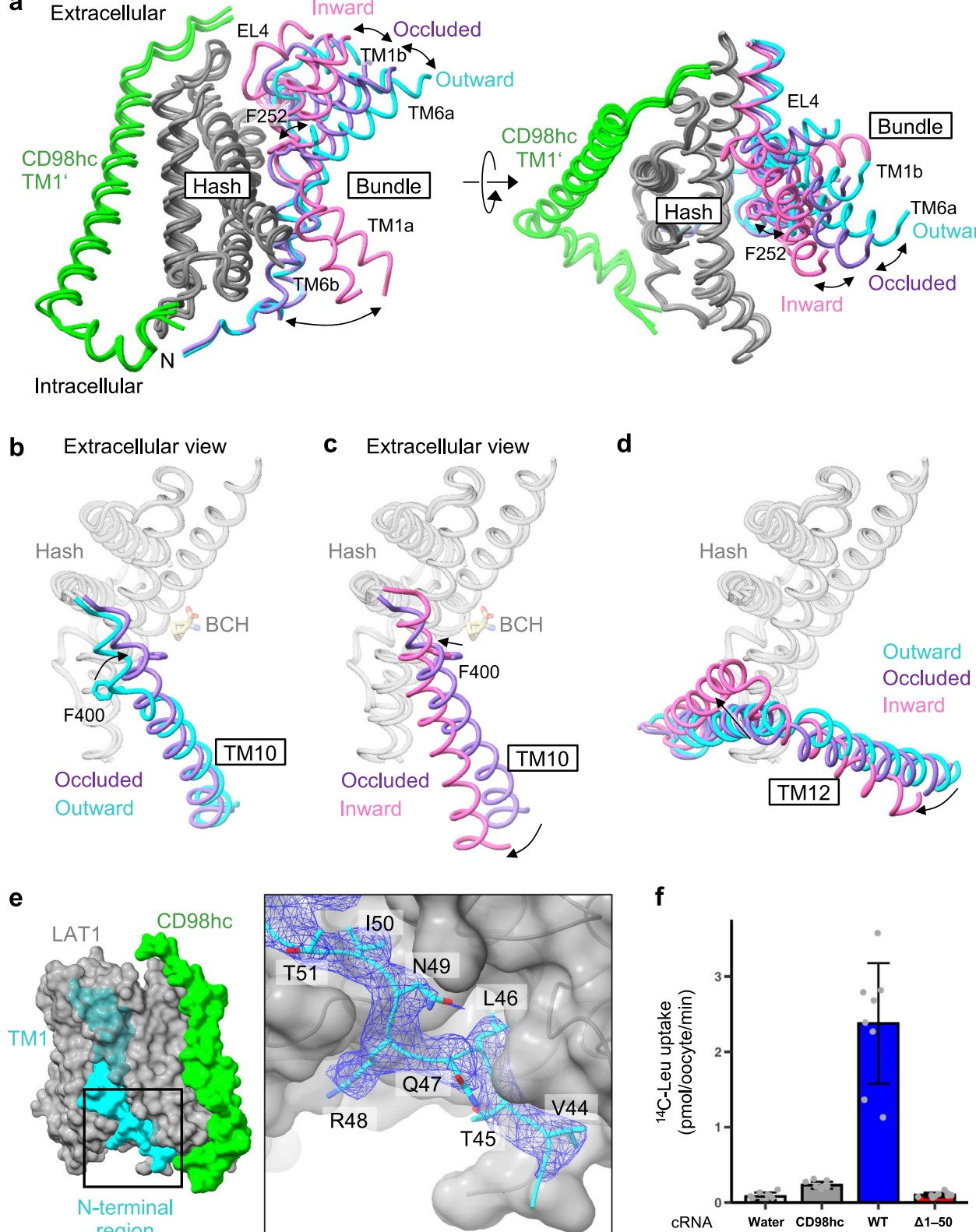

**Fig. 5 | Structural rearrangements of LAT1 for substrate transport.**
**a** Superposition of three major conformational states of LAT1 observed in the study, depicting the rocking-bundle movements relevant to substrate translocation. The hash domain of LAT1 is colored gray and TM1' of CD98hc is green. The bundle domain is colored differently for three conformations: inward-open (pink), occluded (purple) and outward-open (cyan). **b**, **c** Superposition of TM10 in the three conformations. **d** Superposition of TM12 in the three conformations. **e** Close-up view of the N-terminal region of LAT1 in the outward-open state. The cryo-EM density map for the region is shown as blue mesh, contoured at 0.06. **f** Uptake of L-[14C]Leu into *Xenopus* oocytes expressing CD98hc and wild-type or Δ1–50 LAT1. Data are mean ± SD and each data point represents a single oocyte (*n* = 8 for WT; *n* = 9 water; and *n* = 10 for CD98hc and Δ1–50). Source data are available with this paper as Source Data file.

cholesterol bound in the cleft between TM9 and TM12 shows prominent density only in the inward-open state and gets blurred towards the occluded and outward-facing states, suggesting its conformation-specific binding. Indeed, TM12 moves drastically during the transporter's conformational change (Fig. 5d) and generates a cleft suitable for cholesterol binding only in the inward-facing state (Supplementary Fig. 7c). This transient binding of cholesterol to the TM9-TM12 cleft may explain why cholesterol is required to enhance the activity of LAT1 in vitro[21,22], although the putative binding site predicted by the homology model is different from that observed here.

## Discussion

In this study, by combining the lipid nanodisc system and cryo-EM, we have resolved the structures of LAT1 bound to different compounds in three major conformations, outward-facing, occluded and inward-facing, delineating the transport cycle and the structural changes associated with inhibition (Fig. 6 and Supplementary Fig. 8). One significant finding is the first atomic insight into the action of JPH203, a LAT1 inhibitor currently in human clinical trials for cancer treatment[17,18]. Previous pharmacophore modeling suggested that the substrate-binding pocket of LAT1 has a large hydrophobic "free" space capable of accommodating the hydrophobic side groups of inhibitors or prodrugs[3,35]. Our structural data suggest that this long-predicted hydrophobic space is not a rigid, pre-defined pocket. Instead, it consists of highly mobile structural elements, such as TM6, TM10 and part of TM3, which dynamically adapt their conformations to fit the ligands of varying shapes. JPH203 exploits this dynamic nature of the pocket, wedging its amino-phenylbenzoxazol moiety into a flexible segment of TM10. This interaction, in turn, expands the hydrophobic space and prevents the rotation of Phe400 into the substrate-binding pocket, thereby arresting LAT1 in an outward-facing state (Fig. 6a). JX inhibitors largely occupy the same hydrophobic pocket, with only JX-119 interacting extensively with TM3 by adopting a U-shaped conformer similar to JPH203 (Supplementary Fig. 3c). The fact that JX-075 and JX-078 do not show extensive interaction with TM3 but still show comparable inhibition suggests that the interaction with TM10 may be the main determinant of the inhibitory effects.

We have also elucidated the structural mechanism by which the classical system L inhibitor BCH[36,37] acts as a genuine substrate that induces the antiport. The interaction of the norbornane moiety of BCH with LAT1 induces the rotation of Phe400 into the substrate-binding site and a tightening of the hydrophobic pocket (Fig. 6b, c), which is a structural change involved in the transport cycle and is in stark contrast to the pocket expansion by JPH203. Such contrasting effects highlight the structural flexibility of LAT1 in ligand recognition and can be exploited to strategically design compounds with better

transportability or, conversely, stronger inhibition. Our study has also revealed the recognition mechanism of melphalan, a chemotherapeutic agent that is known to be slowly transported by system L[3]. Notably, the occluded pocket observed for BCH is too narrow to accommodate melphalan. This is consistent with the observed slower transport rate of melphalan and other large amino acid derivatives such as $T_3$. The transport process of these large amino acid derivatives may involve a "loose" occluded state, where the substrate is not as tightly confined by surrounding residues as it is for BCH, yet it could still be translocated across the membrane by a series of conformational changes. Another possibility is that the previously proposed "distal pocket" plays a role in the transport of such larger substrates.

Previous studies have explored systematic substitutions on Phe or Tyr to generate high-affinity inhibitors or pro-drugs with improved transport through LAT1[38]. In general, *meta*-substitutions with large hydrophobic moieties are known to increase the affinity but decrease the transport rate, making them inhibitors rather than substrates[38]. From our structure of JPH203, we can deduce that the *meta*-substituted moieties would project toward TM10 (Fig. 2a) and prevent the helix rotation at Phe400, thereby exerting inhibitory effects. Even with the same *meta*-substitution, a longer methylene linker can sometimes increase the transport rate, as shown for valproic acid conjugates[39,40]. The flexibility of the longer linker may allow the side group to avoid severe steric hindrance against TM10, presumably by shifting it towards TM3 or TM6, thus allowing transport.

The O-linked conjugation on Phe (i.e., Tyr derivatives) has proven to be a useful strategy for generating high-affinity inhibitors for system L[41]. While *para*-O-linked substitution has yielded LAT1-selective inhibitors (e.g. JPH203 and SKN-103)[42], *meta*-O-linked substitution has mostly yielded non-selective inhibitors for both LAT1 and LAT2 (e.g. JX-009 and KYT-0283)[26]. Although this can be partially explained by our structure of JPH203, where its *para* substituent is well positioned in the hydrophobic pocket (Fig. 2a), the three JX inhibitors mentioned above (JX-075, JX-078 and JX-119) are *meta*-O-linked but still place their side group roughly in the same pocket (Supplementary Fig. 3a), suggesting that the linkage position cannot fully account for the selectivity in the case of O-linked conjugates. It appears that the cyclized core of the JX inhibitors has a slightly different pose than the non-cyclic core of JPH203 (Supplementary Fig. 3b, c), which may help to fine-tune the position of the *meta*-substituent for optimal fit. However, given that the LAT1/LAT2 selectivity of these JX inhibitors is not yet known, the critical determinants of LAT1 specificity remain to be investigated.

Our observation that all inward-open particles obtained under different ligand conditions were in the apo state suggests that this inward-open state disfavors ligand binding (Fig. 6d). This is consistent with the previously reported functional asymmetry of LAT1[43], where

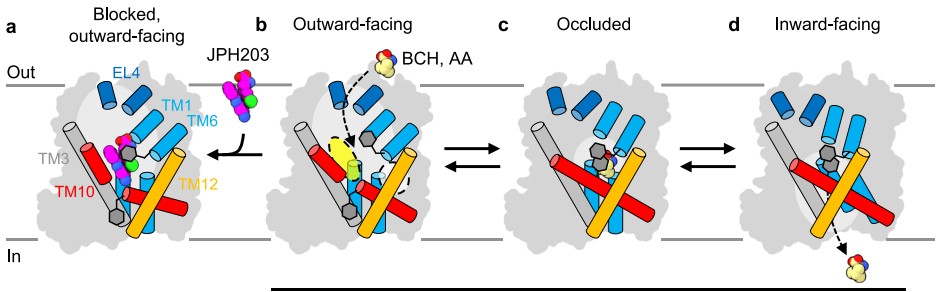

**Fig. 6 | Transport cycle of LAT1 and inhibition mechanisms. a–d** Schematic representation of amino acid transport by LAT1 and its inhibition by JPH203 and BCH. All four panels represent experimental structures reported in the study. In the outward-facing state (**a, b**), LAT1 can accept substrates from extracellular solvent, with mobile Phe252 on TM6 facilitating access (**b**). The white transparent circle depicts the distal pocket predicted previously, and the yellow circle the hydrophobic space found in this study. When JPH203 binds (**a**), TM10 is bent, Phe252 closes, and TM1b and TM6a are pushed open, locking the transporter in the outward-facing state and blocking substrate access. Binding of BCH induces the occluded conformation (**c**), characterized by the rotation of Phe400 on TM10 toward the pocket. Eventually LAT1 transitions to the inward-facing state (**d**), which has low affinity for the substrate and releases it into the cell.

intracellular substrates have lower apparent affinity than extracellular substrates, with an estimated ~1000-fold difference in the Michaelis-Menten constant. Our apo outward-open structure shows that Phe252 is flipped upward, and the hydrophobic pocket is more widely exposed to the solvent (Figs. 3b, 6b), ready to accept extracellular substrates. This asymmetric mechanism allows LAT1 to control the intracellular pool of amino acids against a relatively lower pool of extracellular amino acids, supporting its role as a harmonizer in the context of amino acid homeostasis[44].

Two recent independent studies reported the binding of lipid molecules to LAT1 using computational[45] and mass spectrometry analyses[46]. The molecular dynamics simulations and docking analyses showed that one cholesterol molecule bound in the cleft between TM9 and TM12 (CHOL3 in Supplementary Fig. 7c) is the most stably bound among other species in the inward-facing structure, consistent with our observation. The mass spectrometry revealed that, among several lipid species co-purified with LAT1–CD98hc, phospholipids are stably bound to the heterodimer, whereas cholesterol is lost upon introduction into the gas phase[46]. These results and our observation highlight the presence of persistent and transient lipids, which are involved in the complex stabilization and the conformational change, respectively. The lipid-exposed, transiently formed pockets could be a potential target for allosteric modulation of the transporter.

In conclusion, the structures of LAT1 bound with substrates and inhibitors provide a basis for understanding its amino acid transport and inhibition mechanisms, and establish a blueprint for the improved design of transportable or non-transportable drugs targeting system L. Additionally, the nanodisc system employed here holds promise for further structural and biochemical characterization of LAT1.

## Methods

### Ethics statement
The animal experiments conformed to the guidelines outlined in the Guide for the Care and Use of Laboratory Animals of Japan and were approved by the Chiba University Animal Care Committee (approval number. 30-174), and performed according to ARRIVE guidelines.

### Protein expression and purification
Human LAT1–CD98hc was expressed and purified essentially as described[23]. To improve structural integrity of the protein, we modified purification protocols in several ways. First, we used high-expression batches of HEK293 cells, by using concentrated baculoviruses for maximal infection efficiency of two components[47]. Second, we skipped a FLAG-tag purification step and used only the GFP-nanotrap for affinity purification. Third, the wash buffer for GFP-nanotrap was supplemented with phospholipid mixture (0.0025% POPC, 0.0025% POPG (w/w)). Finally, gel filtration in the presence of detergent was skipped, and the GFP-nanotrap eluate was directly concentrated and used for nanodisc reconstitution. For nanodisc reconstitution, purified protein was mixed with POPC:POPG:cholesterol (2:2:1 (w/w)) and MSP2N2 at a 1:5:100 molar ratio and incubated overnight at 4 °C, with the step-wise addition of Bio-Beads SM-2 as described[48]. The reconstituted complex was further purified by size-exclusion chromatography (SEC) on a Superose 6 Increase 3.2/300 column with the SEC buffer (20 mM Tris-HCl, pH 9.0, 150 mM NaCl).

### Monoclonal antibody generation and selection
4-week-old female MRL/MpJJmsSlc-*lpr/lpr* mice were purchased from Nihon SLC (Shizuoka, Japan). The temperature of the animal breeding room was controlled at 20–25 °C, and the lighting was alternated between day and night at 12 h/12 h. Mice were immunized with liposomes made of purified LAT1–CD98hc and Egg PC (Avanti Polar Lipids, Inc., Birmingham, AL.). After several rounds of immunization, splenocytes from the immunized mice were fused with P3U1 myeloma cells and generated hybridomas. Conformational and extracellular-domain recognizing antibodies produced by hybridomas were screened by using liposome-ELISA, antigen-denatured-ELISA[49] and flow cytometry. IgGs from established clones were purified by Protein G column (Cytiva, Inc., Marlborough, MA.) from culture supernatants and digested into Fab fragments by Papain (Nacalai tesque, Inc., Kyoto, Japan), and further purified on Protein A column (Cytiva). A total of 20 clones were assessed for their binding sites on LAT1–CD98hc using negative-stain microscopy. The Fab fragment from clone 170 (Fab170), which showed rigid binding on the extracellular epitope of CD98hc, similar to a previously characterized MEM-108[23], was used for the structural studies.

### Cryo-EM sample preparation
Nanodisc-reconstituted LAT1–CD98hc was complexed with Fab170, subjected to gel filtration and concentrated to about 15 mg/ml for cryo-EM sample preparation. 3 μl of sample was applied to glow-discharged Quantifoil or C-flat holey carbon grids (copper, 400 mesh, 1.2/1.3 hole size) and blotted for 3–4 seconds using Vitrobot Mark I. To improve particle distribution, 1.5 mM fluorinated Fos-Choline-8 was added immediately before sample application. For JPH203, the inhibitor was added from a 2 mM stock solution in DMSO to a final inhibitor concentration of 20 μM and the mixture was further incubated at 37 °C for 10 min to ensure full binding. For L-Phe (final 5 mM), melphalan (final 400 μM), BCH (final 30 mM) and $T_3$ (final 50 μM), the ligands were added to the sample approximately 1 hour before vitrification and kept on ice. For $T_3$, we also tried longer incubation, by including $T_3$ before the final gel filtration step and all subsequent steps, but neither of the strategies yielded $T_3$-bound LAT1 structures.

### Cryo-EM data acquisition, processing and structure determination
A total of 84,295 cryo-EM movies were collected on multiple separate sessions, summarized in Supplementary Fig. 2e–k. All data were collected on the same Titan Krios G3 microscope equipped with a Bio-Quantum K3 camera at a nominal magnification of 105 k×, which corresponds to a calibrated pixel size of 0.837 Å /pix. The camera was operated in the counted super-resolution mode with a binning factor of 2 and the energy filter slit width was set to 30 eV. The electron flux rate was 15 e$^-$/pix/sec, the exposure time 2.5 sec, and the total exposure 51 e$^-$/Å$^2$. All movies were fractionated into 50 frames. Data were collected with EPU software with the aberration-free image shift method. Data quality was monitored with cryoSPARC Live v2.15[50].

All cryo-EM data processing was performed in RELION 3.1[51,52]. Movies were motion-corrected in MotionCor2[53] with 5 × 5 patches and the contrast transfer function (CTF) was estimated in CTFFIND4[54]. Particles were picked with a Laplacian-of-Gaussian picker and Topaz[55] and extracted with down-sampling to a pixel size of 3.45 Å. All good particles after rounds of 2D and 3D classifications were combined, duplicates removed, and the non-redundant particles were extracted with a final pixel size of 1.5345 Å. Further 3D classification showed that the L-Phe, JPH203, $T_3$ (long incubation) and no-substrate datasets contained only one conformation, whereas the $T_3$ (short incubation) and BCH datasets contained multiple conformations (Supplementary Fig. 2h, i). Focused classification showed that these datasets contained the apo inward-open structures to a varying ratio, which were further classified and combined to yield a single reconstruction (Supplementary Fig. 2l). The $T_3$ (short incubation) dataset, after excluding the inward-open particles, yielded the apo outward-open structure and showed no indication of $T_3$ binding (Fig. 3e and Supplementary Figs. 2h, 5). The no substrate and $T_3$ (long incubation) datasets contained only the apo inward-open particles (Supplementary Figs. 2j, k, 5). After combining the particles of the same conformation, Bayesian polishing[56] was performed with trained parameters.

Even after separating the discrete conformational changes, the continuous structural flexibility between each component (LAT1,

CD98hc, Fab and the surrounding nanodisc) hampered high-resolution reconstruction for the region of interest. To overcome this, we resorted to multibody refinement[57]. Details of the refinement parameters and particle selection strategies are described elsewhere (manuscript in preparation). Briefly, three bodies were defined, designated as the "core", "Fab" and "TMD" (Supplementary Fig. 2d). The "core" mask includes LAT1–CD98hc excluding the nanodisc and Fab170, the "Fab" mask includes only Fab170, and the "TMD" mask includes LAT1, TM1' of CD98hc and the entire nanodisc. Particles were selected through repeated "refine, subtract and classify" cycles, and the final reconstructions of the "TMD" body from multibody refinements were used for model building of the TMD for all datasets. The models for the whole complex (LAT1–CD98hc–Fab170) were built only for the JPH203-bound, BCH-bound and inward-open apo states by using the consensus map as representatives of the outward-facing, occluded and inward-facing conformations, respectively.

A published inward-open structure (PDB ID: 6IRS) was first rigid-body fitted into the maps in ChimeraX 1.0[58]. Subsequent model building was performed in COOT 0.9[59] with ligand restraints generated in AceDRG[60]. Interactive refinement and geometry adjustment were performed in ISOLDE[61]. Final refinement was performed in Servalcat[62] using two unfiltered half maps. The BCH compound (Sigma) contains exo- and endo-carboxy isomers with an unknown ratio. We chose an isomer with an exo-carboxylic group for modeling, based on a previous NMR study of BCH from the same supplier[63]. Data collection and refinement statistics are shown in Supplementary Tables 1 and 2.

### Cryo-EM ligand validation

For ligand validation in cryo-EM maps, we utilized map validation tools in REFMAC5[64]. First, we used Servalcat[62] to calculate "omit" Fo – Fc difference maps from the two unfiltered half maps and the ligand-omitted models for each ligand-bound state. The Fo – Fc maps clearly revealed the presence of the ligands (Supplementary Fig. 5). Second, we used EMDA[65] to calculate Fo – Fo maps between the ligand-bound and apo states. Maps were first fitted to each other using FSC-based auto-resolution threshold and providing a "TMD" mask. Then, the apo map was subtracted from each ligand-bound map. This analysis revealed clear densities for all ligands, JPH203, L-Phe, melphalan and BCH, fitting well the models (Supplementary Fig. 5).

### Transport measurements and expression analysis using X. laevis oocytes

Functional and expression analyses of LAT1 using *X. laevis* oocytes were conducted as described previously unless otherwise specifically denoted[23]. LAT1 mutants with amino acid substitutions or an *N*-terminal truncation (Δ1–50) were constructed by whole-plasmid PCR using PrimeSTAR MAX DNA polymerase (Takara). The corresponding codons were altered as follows for amino acid substitution: S144N (AAT), F252Y (TAT), F252W (TGG), Y259F (TTC), F394Y (TAT), F400V (GTG), F400I (ATT), F400L (TTA), and F400W (TGG).

Uptake experiments of LAT1 in *X. laevis* oocytes were performed in Na⁺-free ND96 buffer containing 50 µM of L-[14C]Leu (3.3 Ci mol⁻¹, Moravek) for 30 min at room temperature. The indicated concentrations of JPH203 or BCH were added to the buffer when specified. After the lysis of oocytes with 10% (v/v) SDS, the radioactivity was determined using a β-scintillation counter (LSC-3100, Aloka, Tokyo, Japan). In efflux experiments, oocytes were preinjected with 50 nL of 100 µM L-[14C]Leu (3.3 nCi/ oocyte). After an extensive wash with ice-cold Na⁺-free ND96 buffer, the oocytes were incubated in the buffer with or without the indicated concentrations of JPH203, BCH, or L-Leu for 15 min at room temperature to induce efflux of preloaded L-[14C]Leu. Then the radioactivity in the buffer and the remaining radioactivity in the oocytes were separately counted. L-[14C]Leu efflux was expressed as a percentage of total radioactivity (the radioactivity in the buffer divided by the sum of the radioactivity of the buffer and the remaining radioactivity in oocytes).

Detection of LAT1 by immunoblotting in the total membranes of *X. laevis* oocytes was performed with anti-LAT1 (1:2,000, KE026; TransGenic) and peroxidase goat anti-rabbit IgG (1:10,000, Jackson ImmunoResearch). Immunofluorescence detection of LAT1 in paraffin sections of *X. laevis* oocytes was performed with anti-LAT1 (1:500, J-Pharma) and Alexa Fluor 488-conjugated anti-mouse IgG (A21202, 1:1000, Invitrogen). Images were acquired using a fluorescence microscope (BZ-9000, Keyence) equipped with a ×100 objective lens (CFI Plan Apo λ, numerical aperture 1.40, Nikon). Reproducibility of the results was confirmed by independent experiments using different batches of oocytes.

### Materials for radioactive assays

14C-leucine (338 mCi/mmol) was purchased from Moravek Biochemicals (Brea, CA). Standard amino acids and 2-aminobicyclo[2.2.1]heptane-2-carboxylic acid (BCH) were purchased from Sigma-Aldrich (St Louis, MO). JPH203 ((S)-2-amino-3-(4-((5-amino-2-phenylbenzo [d] oxazol-7-yl)methoxy)-3,5-dichlorophenyl) propanoic acid, CAS No. 1037592–40-7) (2HCl salt; purity > 99%) was provided by J-Pharma Co., Ltd (Tokyo, Japan). Unless otherwise stated, other chemicals were purchased from Wako Pure Chemical Industries (Osaka, Japan).

### Reporting summary

Further information on research design is available in the Nature Portfolio Reporting Summary linked to this article.

### Data availability

Source data are available with this paper as a Source Data file. The atomic coordinates have been deposited to Protein Data Bank under accession numbers 8KDD, 8KDF, 8KDG, 8KDH, 8KDI, 8KDJ, 8KDN, 8KDO and 8KDP. Cryo-EM maps have been deposited to Electron Microscopy Data Bank under accession numbers EMD-37132, EMD-37134, EMD-37135, EMD-37136, EMD-37137, EMD-37138, EMD-37140, EMD-37141 and EMD-37142. The particle image files have been deposited to the Electron Microscopy Public Image Archive under an accession number EMPIAR-12031. All other data can be found in the Supplementary Information file, or will be available upon request.

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

## Acknowledgements

We thank Werner Kühlbrandt for providing research infrastructure and discussions; Deryck J. Mills for microscope management and training; Susann Kaltwasser, Simone Prinz, Mark Linder and Sonja Welsch in the Central Electron Microscopy Facility of the Max Planck Institute of Biophysics for technical assistance in electron microscopy; Sabine Häder, Christina Kunz and Heidi Betz for assistance in laboratory experiments; Juan Castillo, Özkan Yildiz, the Central IT team and the Max Planck Computing and Data Facility for maintaining the computational infrastructure; and J-Pharma for providing JPH203. This work was supported by the Max Planck Society, Kazato Research Foundation, JSPS KAKENHI (21K15031), and The Uehara Memorial Foundation to Y.L.; Basis for Supporting Innovative Drug Discovery and Life Science Research (BINDS) from AMED (JP23ama121013) to T.M.; The Medical Research Council, as part of UK Research and Innovation (MC_UP_A025_1012) to K.Y. and G.M; the Project for Cancer Research and Therapeutic Evolution from AMED (JP23ama221121 and JP24ama221121) to Y.K. Y.L. was supported by Toyobo Biotechnology Foundation Fellowship and Human Frontier Science Program Long-Term Fellowship.

## Author contributions

Y.L., O.N. and Y.K. initiated the project. Y.L. performed EM sample preparation, data collection and structure determination. C.J. and R.O. prepared mutants and performed functional assays. R.O. and M.X. performed immunostaining and western blotting. S.O. and T.M. generated monoclonal antibodies and performed antibody screening. R.W. and G.M. helped with EMDA overlay and Fo – Fo difference map calculation. K.Y. and G.M. assisted with map and model validation using Servalcat. Y.L. and Y.K. wrote the manuscript, with contributions from all co-authors.

## Funding

## Competing interests

Y.L. receives research funding from J-Pharma. O.N. is a co-founder and scientific advisor of Curreio. All the other authors declare no competing interest.
