## [Transparent Peer Review file · Nature Communications]

Structural basis of anticancer drug recognition and amino acid transport by LAT1

Corresponding Author: Dr Yongchan Lee

This manuscript has been previously reviewed at another journal that is not operating a transparent peer review scheme. The manuscript was considered suitable for publication without further review at Nature Communications.

Version 0:

Reviewer comments:

Reviewer #1

(Remarks to the Author)

The study by Lee et al. report the structure and characterization of LAT1 inhibition by several compounds. The authors determine structures of the protein in multiple states and compound-bound complexes. The topic itself is important and significant experiments appear well-designed, and extensive data was collected to address their hypotheses.

However, there are inconsistencies and concerns in the manuscript, both technical and in presentation. Therefore, this manuscript is not publishable in its current form, and will require a thorough reanalysis of the data.

Critical concerns:

- The refinement resolution appears set significantly higher than the map resolution, which will cause overfitting. A more appropriate resolution limit for refinement, likely the overall map resolution, should be used.
- The model statistics for many of the models are unacceptable. This includes the RMS deviations for bond lengths and angles, clash score, and rotamer outliers. This could be due to an inappropriate refinement resolution (noted above) and/or incorrect weighting of the geometric restraints.
- As written, it is very difficult to understand the origins of the “apo” map. As a first reading, this map is a serendipitous output of the T3 datasets. However, table S1 clearly shows a “No substrate” dataset was collected. Clarity here is critical to understanding how the difference maps were calculated in Fig S1 and S5. Furthermore, the “No substrate” dataset should be used throughout as the genuine “apo” map, and to validate empty binding sites of the T3 maps.
- Relatedly, in tables S1 and S2, it is very difficult to follow which datasets contributed to various maps and models. These should be explicitly labeled.
- The authors argue that the T3-bound and inward-facing structures are apo, but no direct experimental evidence is shown for this. Panels should be included in Figure S5 for this data.
- More generally, panels should be included in S5 for all structures. Also, all conditions should be shown on an equivalent scale, which can be done by scaling all maps on a separate, unchanging portion of the model.

Major concerns:

- There are two pink models in Fig 5a. However, only one apo model was described in the text. Where does this second model come from?
- The authors use an enticing and novel strategy to generate the apo, inward-open map using particles from several separate conditions. While this makes intuitive sense, the number of particles from each originating dataset in the input and final reconstruction should be recorded to identify any bias in the classification.

Minor concerns:

- Terming BCH as a “facilitator” of transporter turnover is incorrect, as it acts orthosterically. More precisely, it is a competitive substrate.
- Figure S2e is missing a panel.
- Figure S4 should be rescaled to show the residual activity of S144N/F400V. At present, the activity of this mutant appears

similar to the negative control conditions.

- Figure 4h is missing an inset showing export expressed as a percentage of total radioactivity, as described in the figure legend.
- The authors should show density for the noted ordered N-terminal domain.
- The authors use a novel strategy for generating the inward-facing apo map by combining particles from multiple datasets. To enable future development of this strategy, the separate stacks of input particles from each condition, the final particle stack, and the final refinement .star file, should be deposited into EMPIAR.
- In Fig. S2g, the label "Inward #1" appears incorrectly to indicate these particles come from the global alignment, rather than the focused refinement of the TMD, as is used for all other inward particle sets. This should be corrected.
- Focused refinement of the TMD often yielded a worse overall resolution than consensus refinement. While I suspect this strategy might have yielded improved resolution the TMD, this should be shown as a local resolution map of the entire complex for at-least one dataset to validate the methodology.

Reviewer #2

(Remarks to the Author)

NCOMMS-24-01035-T

Structural basis of anticancer drug recognition and amino acid transport by LAT1

Yongchan Lee, Chunhuan Jin, Ryuichi Ohgaki, Minhui Xu, Satoshi Ogasawara, Rangana Warshamanage, Keitaro Yamashita, Garib Murshudov, Osamu Nureki, Takeshi Murata, Yoshikatsu Kanai

General.

Please do not mistake my brevity for lack of detailed analysis nor enthusiasm.

This is an elegant and very informative study not only for LAT1 specifically but also for further characterization of inhibitors vs transported substrates. Many investigators, particularly in the cancer therapeutic realm, do not even entertain the idea that transporters, channels and pumps are ALL bidirectional under certain conditions – sometimes non-physiologic while in many cases these states may be physiological depending on the cell type and tissue. Importantly, as summarized by the authors, their structures "...demonstrate that this long-predicted hydrophobic space (within human LAT1) is not a rigid, pre-defined pocket."

Below are some additional thoughts to increase the appeal and readability of the MS.

Major.

Data within main body of MS

1. (p.6, line 131-140): Figure S4 is important for the results and should be in the main body of the MS
2. (Structural rearrangements in the transport mechanism): Some summary Figure representation of the (a) inward-facing, (b) outward-facing, and (c) occluded should be part of the Main MS. This Summary will help drive home the point that the authors are able to structurally differentiate these DIFFERENT LAT1 states. This is the explicit demonstration of the LAT1 transport cycle observe in the real structure. Perhaps something similar to Fig S7.

Minor.

- (li 184): Confusing due to "...and that of JPH203...": in Fig 3f,g. This shows l-Phe and melphalan
- (li 189-191): Fig 4g does not show interaction with TM3. The "spaces" shown in Figs 4b-d indicate that TM3 is part of this cavity would be good to have a panel better illustrating this point
- (li 194): binding of Term Cl to Y259 is Fig 3g
- (li 242): NO fig 4i
- (li 251): Better explanation of "inward-open structures (Fig. S2g,h,i,j)..." is needed. While I appreciate the labeling in Fig S2, the multiple labels per panel make it unclear whether the authors intend the overall state or the "Inward#" in their text description.
- (li 261): showing Phe252 in Fig 5a can help further clarify states
- (294): "Discussion"

Reviewer #3

(Remarks to the Author)

The manuscript by Lee et al is a very interesting and extensive work on a transporter (LAT1) that is a hot spot for both biochemistry of membrane proteins and relevance to human health. The manuscript may potentially give important new information on this transporter and could represent a milestone for drug discovery and design. However, authors give a limited overview of previous knowledge on LAT1, make some confusion on the transporter specificity and distinction between substrates and inhibitors. Therefore, they should make an extensive effort in improving interpretation and discussion of results also in the light of previous knowledge.

introduction.

Line 48. At present it is not convenient starting a manuscript with an old classification. If authors like to remark "System L" they should also include information on the SLC classification and cite that LAT3 and LAT4 belong to a different family (SLC43) with respect to that of LAT1 and 2 (SLC7). Moreover, SLC43 family members do not form heterodimers with SLC3 members, while SLC7A5 and A8 do.

Lines 48-52 and Fig 1:

The known transported amino acid should be detailed. LAT1 mainly transports essential amino acids including the basic amino acid histidine as well as the non-essential cysteine and tyrosine (see several papers and reviews on LAT1 and SLC7 family transporters). Indeed, it also transports some drugs but not JPH203, as the authors themselves declare in the manuscript; then, Fig. 1b should be updated in line with the text: the actual substrates and the transported drugs (not JPH203) should be depicted.

Fig. 1a should be removed since it does not give any specific information on LAT1.

Lines 57-58.

LAT1 is first responsible for transport of essential amino acids through the BBB, besides of drugs. This should be remarked with appropriate refs.

Line 69.

BCH is a substrate that also behaves as inhibitor by competing with substrates. This was already described in 2013 (Geier et al. PNAS, ref 21 of this manuscript) by an experiment mostly identical to that described by the authors.

Results:

Line 98. More detailed information on cholesterol effects on LAT1 have been previously described and should be reported (see Cosco et al Sci. Rep. 2020).

Lines 136-140.

The activity of the mutant S144N in transfected oocytes is lower than the activity of the sole CD98 transfected oocytes (S4b). The activity of the double mutant is comparable (or only slightly higher) to that of the single S144 and cannot be used to gain correct information on inhibition. Therefore, the conclusion concerning the selectivity LAT1/2 is not appropriate. Moreover, the mutant F400V that is the only one showing a measurable activity exhibits a behavior with the inhibitor equal to WT (see fig. S4c). Then, how do the authors justify the presence of residual transport activity in the double mutant with respect to the single S144? This part needs to be removed or reconsidered.

Line 181.

there is something wrong with the figure number. Flipped up F is visible in 3h not in 3d.

Lines 199-206

Authors describe alkylating agents, but they only mention mustard. Indeed, LAT1 also interacts with other alkylating agents. Therefore, unless authors have more than one example, PGA should be treated as a single case study or other data from literature should be cited.

Line 208.

The role of BCH as a substrate and, therefore, as a competitive inhibitor (see line 69 of introduction) is not an original data (see point above). This needs to be well clarified also in the view of the relationships with the structure. Fig. 4 h should be modified; since BCH is a substrate it induces the normal antiport function not a trans-stimulation phenomenon. The manuscript will benefit from a good effort in discussing the basis for transport of this compound in relationships to the occluded conformation (they only try on line 231). Is it a fully occluded state, or is it occluded but oriented toward a side of the membrane etc? An appropriate interpretation/discussion of data would be of great interest for readers.

Lines 234-236.

Again, this part should be rewritten on the basis of the finding that BCH is a substrate.

Line 243.

This is true, but not an original result, please cite the appropriate previous findings (see above).

Lines 248-253.

Is it possible that this state represents the more stable conformation, a sort of ground state? Authors completely ignore previous findings on the functional asymmetry of LAT1. Previous literature and specific reviews on LAT1 on this topic could be very useful for a more interesting and appropriate discussion of the reported 3D data.

Lines 278-283.

The finding that the protein lacking 1-50 residues is not active suggests that this portion of LAT1 is relevant to the transport mechanism or its regulation. Are there PTM sites? This issue should be better discussed and clarified.

Lines 291-292.

Please clarify this part also in view of previous literature data on cholesterol interactions.

Discussion

The discussion section is poor being a list of the main findings. Indeed, this section does not add any new information or speculation/hypotheses about the results. It should be rewritten also considering the observations on Results.

Specific comments:

Lines 299-300.

Please discuss the similarities/advancements with respect to the previously 3D structures and previous published computational predictions of LAT1 and interactions with substrates and inhibitors.

Line 303.

If BCH is a substrate, as it is, it does not affect the transporter turnover (it is not an allosteric activator). It just induces the antiport function.

Version 1:

Reviewer comments:

Reviewer #1

(Remarks to the Author)

The study by Lee et al. report the structure and characterization of LAT1 inhibition by several compounds. The authors determine structures of the protein in multiple states and compound-bound complexes. The topic itself is important and significant experiments appear well-designed, and extensive data was collected to address their hypotheses.

The authors have done a commendable job resolving my technical and scientific concerns I had in the previous review. I have no more concerns and am happy for this paper to proceed to publication.

Reviewer #3

(Remarks to the Author)

The responses of the authors to my concerns were appropriate and the manuscript has been changed accordingly. Only one issue should be dealt with more extensively, that is the double mutant issue (now lines 151-155). The results should be better explained. Some comments, based on the author hypothesis, should be added in the discussion or in the results section. After this change the manuscript can be acceptable.

Response letter to the reviewers

Reviewer #1 (Remarks to the Author):

The study by Lee et al. report the structure and characterization of LAT1 inhibition by several compounds. The authors determine structures of the protein in multiple states and compound-bound complexes. The topic itself is important and significant experiments appear well-designed, and extensive data was collected to address their hypotheses.

However, there are inconsistencies and concerns in the manuscript, both technical and in presentation. Therefore, this manuscript is not publishable in its current form, and will require a thorough reanalysis of the data.

Answer:

We appreciate the reviewer's critical assessment of our work and raising important concerns on the technical and presentation aspects. We have now undertaken a thorough reanalysis of the data and have corrected figures and texts for clearer presentation. Specific point-by-point responses are provided below.

Critical concerns:

- The refinement resolution appears set significantly higher than the map resolution, which will cause overfitting. A more appropriate resolution limit for refinement, likely the overall map resolution, should be used.

Answer:

As detailed in our next response, we have revised the model refinement procedure, which have now yielded significantly improved statistics. To clarify our criteria for choosing the refinement resolution limit, the recommendation by the REFMAC5 developers is that the refinement resolution should NOT be cut off at the global resolution, because some regions have better local resolution than the global resolution, and therefore setting a hard cutoff would result in a loss of information in these higher-resolution regions. In our particular case, it should be noted that the FSC values were calculated including the lipid nanodisc region (lipids + scaffold), which reduces the average resolution. As can be seen in Fig. S2e-i,1, the local resolutions in the ligand binding site are significantly better. We have set the refinement resolutions close to, but not better than, these best local resolutions. Another note is that even if the user does not set an arbitrary cutoff value, *servalcat* estimates the noise levels of each Fourier components from the half maps and these estimates are used for Fourier map weighting, resulting in down-weighting of higher resolution information.

- The model statistics for many of the models are unacceptable. This includes the RMS deviations for bond lengths and angles, clash score, and rotamer outliers. This could be due to an inappropriate refinement resolution (noted above) and/or incorrect weighting of the geometric restraints.

Answer:

Thank you for pointing out the poor model statistics. First of all, we noticed that the previous RMS deviation value for the BCH consensus model was mistakenly reported as 0.177, which should have been 0.0177. Based on this suggestion, we investigated the cause of the poor model statistics and found that the models coming out of Coot had poor geometry, probably due to too much weight on the data, and this was not removed by 30 iterations of jelly-body refinement in REFMAC5. To overcome this, we interactively corrected all models in ISOLDE, fixing ramachandran outliers, rotamer outliers and severe clashes, and ensuring consistency across all the models. We then refined the models in REFMAC5 using auto-optimized weights, which resulted in significantly improved models. We believe that the new model statistics meet the criteria generally accepted in the field.

- As written, it is very difficult to understand the origins of the “apo” map. As a first reading, this map is a serendipitous output of the T3 datasets. However, table S1 clearly shows a “No substrate” dataset was collected. Clarity here is critical to understanding how the difference maps were calculated in Fig S1 and S5. Furthermore, the “No substrate” dataset should be used throughout as the genuine “apo” map, and to validate empty binding sites of the T3 maps.

Answer:

We acknowledge the confusing notation for the "apo" map in our original manuscript. We would like to clarify that the "T3 short incubation" dataset yielded the "outward-facing" apo map, whereas the "no substrate" data yielded the "inward-facing" apo map. Since all the ligand-bound maps are "outward-facing" (except for BCH), we used the corresponding outward-facing apo map from "T3 short incubation" as a reference map for calculating the Fo-Fo map. To show that the binding site of the T3 map is indeed empty, we have now added an Fo-Fc map for the T3 map in Fig S5. We have also added these descriptions in the Methods section.

- Relatedly, in tables S1 and S2, it is very difficult to follow which datasets contributed to various maps and models. These should be explicitly labeled.

Answer:

Thank you for your suggestion to improve the clarity of Tables S1 and S2. The original datasets that contributed to various maps and models are now explicitly labeled in Tables S1 and S2.

- The authors argue that the T3-bound and inward-facing structures are apo, but no direct experimental evidence is shown for this. Panels should be included in Figure S5 for this data.

Answer:

The cryo-EM densities for apo outward-open (T3 short incubated) and apo inward-open (no-substrate data and combined data) are now shown in Figure S5.

- More generally, panels should be included in S5 for all structures. Also, all conditions should be shown on an equivalent scale, which can be done by scaling all maps on a separate, unchanging portion of the model.

Answer:

We have added equivalent Fo-Fc maps for the two apo maps (outward-open and inward-open). Regarding map scaling, we would like to note that the difference maps are already normalized within the same mask, which is now explicitly described in the figure legends. We have also changed the map contours to the same value ($\pm 6\sigma$) for all maps.

Major concerns:

- There are two pink models in Fig 5a. However, only one apo model was described in the text. Where does this second model come from?

Answer:

We apologize for the unclear illustration. There is only one pink model in Figure 5a. It was probably confusing because there were two identical labels shown in one panel. We have rearranged the labels so that it is clear that one model is shown for each conformation (inward, occluded and outward).

- The authors use an enticing and novel strategy to generate the apo, inward-open map using particles from several separate conditions. While this makes intuitive sense, the number of particles from each originating dataset in the input and final reconstruction should be recorded to identify any bias in the classification.

Answer:

Thanks for the suggestion. We have added the number of particles for the apo inward-open map before and after the "cleaning" 3D classification in Figure S2I. The comparison shows that there is small bias towards selecting the "no substrate" and "T3 long incubation" datasets after the 3D classification, which is probably because these datasets were more homogeneous than the "T3 short incubation" and "BCH" datasets, which contained outward-facing particles in the initially picked particle sets.

Minor concerns:

- Terming BCH as a "facilitator" of transporter turnover is incorrect, as it acts orthosterically. More precisely, it is a competitive substrate.

Answer:

We have revised the misleading notation of BCH being a "facilitator" throughout the manuscript. Where appropriate, we have corrected words to "induce" the antiport. We also avoided the word "stimulate" such as "*trans* stimulation", as this could indeed be misinterpreted as BCH being an allosteric modulator.

- Figure S2e is missing a panel.

Answer:

Thank you for pointing this out. Although we have not been able to reproduce the issue, there may have been a technical issue with the previous PDF file. We have re-created the file and believe the issue has been resolved.

- Figure S4 should be rescaled to show the residual activity of S144N/F400V. At present, the activity of this mutant appears similar to the negative control conditions.

Answer:

We have added a rescaled version of Figure S4a, which shows a significant residual activity of S144N/F400V. We note that the negative control here is the cells injected with water, not with CD98hc cRNA, because CD98hc is known to induce low internal amino acid transport activity in *Xenopus* oocytes under our assay conditions, presumably by forming a complex with intrinsic SLC7 members [for example, [https://doi.org/10.1016/S0021-9258\(19\)49531-8](https://doi.org/10.1016/S0021-9258(19)49531-8)]. Nevertheless, we reproducibly observed significant residual activity in S144N/F400V, and this activity was insensitive to JPH203.

- Figure 4h is missing an inset showing export expressed as a percentage of total radioactivity, as described in the figure legend.

Answer:

We apologize for the inconsistency in the original Figure 4h and its legend. We have now removed the legend pointing that refers to a small inset that was edited out when the original manuscript was finalized.

- The authors should show density for the noted ordered N-terminal domain.

Answer:

We have added the cryo-EM density for the ordered N-terminal domain.

- The authors use a novel strategy for generating the inward-facing apo map by combining particles from multiple datasets. To enable future development of this strategy, the separate stacks of input particles from each condition, the final particle stack, and the final refinement .star file, should be deposited into EMPIAR.

Answer:

Thank you for the very good suggestion. We have deposited the particle image files along with all RELION metadata into EMPIAR, as "Cryo-EM structures of LAT1-CD98hc in nanodisc" with the accession number EMPIAR-12031.

- In Fig. S2g, the label "Inward #1" appears incorrectly to indicate these particles come from the global alignment, rather than the focused refinement of the TMD, as is used for all other inward particle sets. This should be corrected.

Answer:

The reviewer is actually correct in that the separation of "inward subset #1" particles started with the global refinement for the "T3 short incubation" dataset. Although this global classification worked well to separate the discrete conformations (inward vs outward), some additional rounds of classification were needed to clean the particles. These "cleaning" 3D classifications are now indicated by the updated labels "Core focus 3DC" and "TM focus 3DC" above an arrow in Figure S2g for clarity.

- Focused refinement of the TMD often yielded a worse overall resolution than consensus refinement. While I suspect this strategy might have yielded improved resolution the TMD, this should be shown as a local resolution map of the entire complex for at-least one dataset to validate the methodology.

Answer:

The reviewer is correct that the focused refinement of the TMD resulted in better local resolution for the region of interest. To illustrate this point, we have added Figure S2p showing the local resolutions of the JPH203 dataset before and after the multibody refinement.

Reviewer #2 (Remarks to the Author):

NCOMMS-24-01035-T

Structural basis of anticancer drug recognition and amino acid transport by LAT1

Yongchan Lee, Chunhuan Jin, Ryuichi Ohgaki, Minhui Xu, Satoshi Ogasawara, Rangana Warshamanage, Keitaro Yamashita, Garib Murshudov, Osamu Nureki, Takeshi Murata, Yoshikatsu Kanai

General.

Please do not mistake my brevity for lack of detailed analysis nor enthusiasm.

This is an elegant and very informative study not only for LAT1 specifically but also for further characterization of inhibitors vs transported substrates. Many investigators, particularly in the cancer therapeutic realm, do not even entertain the idea that transporters, channels and pumps are ALL bidirectional under certain conditions – sometimes non-physiologic while in many cases these states may be physiological depending on the cell type and tissue. Importantly, as summarized by the

authors, their structures “...demonstrate that this long-predicted hydrophobic space (within human LAT1) is not a rigid, pre-defined pocket.”

Answer:

Thank you for your careful reading of our manuscript and your appreciation of its scientific implications. We do agree that our data provide important insights for further characterization of inhibitors vs transported substrates.

Below are some additional thoughts to increase the appeal and readability of the MS.

Major:

Data within main body of MS

1. (p.6, line 131-140): Figure S4 is important for the results and should be in the main body of the MS

Answer:

Thank you for your important suggestion. Although we agree that Fig. S4 is important, there are mixed opinions from the reviewers, and we would like to leave it up to the editor whether or not to move it to the main manuscript.

2. (Structural rearrangements in the transport mechanism): Some summary Figure representation of the (a) inward-facing, (b) outward-facing, and (c) occluded should be part of the Main MS. This Summary will help drive home the point that the authors are able to structurally differentiate these DIFFERENT LAT1 states. This is the explicit demonstration of the LAT1 transport cycle observe in the real structure. Perhaps something similar to Fig S7.

Answer:

We appreciate this suggestion. Due to a limited space for main figures, we have added the suggested summary figure presentation in Figure S8.

Minor:

• (li 184): Confusing due to “....and that of JPH203...”: in Fig 3f,g. This shows l-Phe and melphalan

Answer:

Thank you for the suggestion. To avoid confusion, we have moved the "(Fig. 3f,g)" to an earlier part of the sentence.

> In the L-Phe- and melphalan-bound structures (Fig. 3f,g).

• (li 189-191): Fig 4g does not show interaction with TM3. The "spaces" shown in Figs 4b-d indicate that TM3 is part of this cavity would be good to have a panel better illustrating this point

Answer:

We have corrected the text to cite Figure 3b,d,g. We have added a new figure panel Fig. 3b, showing a hydrophobic space that accommodates the nitrogen mustard moiety of melphalan.

• (li 194): binding of Term Cl to Y259 is Fig 3g

Answer:

We have corrected the text to cite Fig. 3g.

- (li 242): *NO fig 4i*

Answer:

Thanks for pointing this out. It was probably introduced during our editing process. We removed the legend for the non-existent Fig. 4i.

- (li 251): *Better explanation of “inward-open structures (Fig. S2g,h,i,j)...” is needed. While I appreciate the labeling in Fig S2, the multiple labels per panel make it unclear whether the authors intend the overall state or the “Inward#” in their text description.*

Answer:

We received similar comments about Figure S2 from another reviewer and have now rearranged the figure panels. Specifically, we use the word "inward subset #1" to indicate a particle subset derived from each dataset, and the word "apo inward-open" to indicate the final map obtained after combining these particles.

- (li 261): *showing Phe252 in Fig 5a can help further clarify states*

Answer:

Thank you for the suggestion. We have added Phe252 in Figure 5a to clarify the states.

- (294): *“Discussion”*

Answer:

Corrected.

Reviewer #3 (Remarks to the Author):

The manuscript by Lee et al is a very interesting and extensive work on a transporter (LAT1) that is a hot spot for both biochemistry of membrane proteins and relevance to human health. The manuscript may potentially give important new information on this transporter and could represent a milestone for drug discovery and design. However, authors give a limited overview of previous knowledge on LAT1, make some confusion on the transporter specificity and distinction between substrates and inhibitors. Therefore, they should make an extensive effort in improving interpretation and discussion of results also in the light of previous knowledge.

Answer:

Thank you very much for appreciating the scientific findings presented in our manuscript and their broader implications. Based on the specific comments, we have now extensively modified the text and figures to improve the interpretation and discussion.

introduction.

Line 48. At present it is not convenient starting a manuscript with an old classification. If authors like to remark "System L" they should also include information on the SLC classification and cite that LAT3 and LAT4 belong to a different family (SLC43) with respect to that of LAT1 and 2 (SLC7). Moreover, SLC43 family members do not form heterodimers with SLC3 members, while SLC7A5 and A8 do.

Answer:

Thank you for your suggestion to improve the Introduction. Based on the suggestion, we have rewritten the entire first paragraph of the Introduction, which now starts with the description of LAT1, followed by a traditional "system"-based classification and the modern SLC nomenclature. Following this narrative, the revised text explicitly states that LAT3 and 4 belong to SLC43 and that they do not form heterodimers with SLC3 members.

Lines 48-52 and Fig 1:

The known transported amino acid should be detailed. LAT1 mainly transports essential amino acids including the basic amino acid histidine as well as the non-essential cysteine and tyrosine (see several papers and reviews on LAT1 and SLC7 family transporters). Indeed, it also transports some drugs but not JPH203, as the authors themselves declare in the manuscript; then, Fig. 1b should be updated in line with the text: the actual substrates and the transported drugs (not JPH203) should be depicted.

Answer:

We have added more detailed descriptions for amino acids and drugs transported by LAT1. We have also corrected the figure to reflect this. Due to space limitations, our focus is on the compounds we have used for structural studies, rather than covering the full range of amino acids and drugs transported by LAT1. Therefore, it is worth including the structure of JPH203 in the main figure for clarity. Regarding the distinction between substrate and inhibitor, some compounds are difficult to clearly classify. For example, in the context of cellular amino acid uptake, T3 acts as a strong competitive inhibitor that can be distinguished from amino acid substrates because it induces less efflux of intracellular amino acids [[https://doi.org/10.1016/s0005-2736\(02\)00516-3](https://doi.org/10.1016/s0005-2736(02)00516-3)]. This suggests a low transport capacity for T3. Other studies have implicated LAT1 in T3 transport across the BBB, emphasizing the behavior of T3 as a substrate. Therefore, it is not straightforward to classify each compound as either a substrate or an inhibitor, and for the sake of readability we prefer showing compounds we used for the study, rather than covering known transported drugs.

Fig. 1a should be removed since it does not give any specific information on LAT1.

Answer:

We have removed Fig. 1a.

Lines 57-58.

LAT1 is first responsible for transport of essential amino acids through the BBB, besides of drugs. This should be remarked with appropriate refs.

Answer:

LAT1 indeed plays nutritional roles for the brain. To remark this fact, we have updated the second paragraph to include a reference to Tarlunganu *et al.*, Cell, 2016, who showed that LAT1 knockout mice exhibit symptoms of autism spectrum disorder. We have also revised the sentences to emphasize this in addition to drug delivery to the brain.

Line 69.

*BCH is a substrate that also behaves as inhibitor by competing with substrates. This was already described in 2013 (Geier *et al.* PNAS, ref 21 of this manuscript) by an experiment mostly identical to that described by the authors.*

Answer:

We have revised the text to cite the studies that showed BCH is a substrate. We believe that this fact is underappreciated and many studies often treat BCH and other inhibitors such as JPH203 interchangeably, which is not the case.

Results:

Line 98. More detailed information on cholesterol effects on LAT1 have been previously described and should be reported (see Cosco et al Sci. Rep. 2020).

Answer:

Thank you for pointing out the previous literature that has shown the effects of cholesterol on LAT1. We have added the suggested reference in the indicated sentence. In addition, we have added to the Discussion section two recent papers that have biochemically and computationally analyzed the binding of cholesterol and/or phospholipids to LAT1.

Lines 136-140.

The activity of the mutant S144N in transfected oocytes is lower than the activity of the sole CD98 transfected oocytes (S4b). The activity of the double mutant is comparable (or only slightly higher) to that of the single S144 and cannot be used to gain correct information on inhibition. Therefore, the conclusion concerning the selectivity LAT1/2 is not appropriate. Moreover, the mutant F400V that is the only one showing a measurable activity exhibits a behavior with the inhibitor equal to WT (see fig. S4c). Then, how do the authors justify the presence of residual transport activity in the double mutant with respect to the single S144? This part needs to be removed or reconsidered.

Answer:

We have received similar comments from another reviewer and have now added an enlarged graph of Fig. S4a showing the significant residual activity of the double mutant S144N/F400V. We would like to note that the negative control here is the cells injected with water, not with CD98hc cRNA, because CD98hc is known to induce low but significant amino acid transport in *Xenopus* oocytes in our assay system, presumably by forming a complex with intrinsic SLC7 members [e.g. [https://doi.org/10.1016/S0021-9258\(19\)49531-8](https://doi.org/10.1016/S0021-9258(19)49531-8)]. Notably, when the activity would be completely lost, the uptake would become very close to that with water, as shown for the delta1-50 variant (Figure 5f). This can be explained by the highly expressed mutant LAT1 assembling preferentially with CD98hc compared to intrinsic SLC7 members. Our hypothesis as to why the double variant restores activity compared to the single S144N is that the introduction of the larger Asn is complemented by the conversion of the bulky Phe400 to the smaller Val. This would restore the substrate binding ability in the pocket but abolish the interaction with JPH203. We have supporting evidence from structural studies of LAT2, which is to be published elsewhere.

Line 181.

there is something wrong with the figure number. Flipped up F is visible in 3h not in 3d.

Answer:

Thanks for pointing this out. It appears that there was some corruption in the PDF file, although we have not been able to reproduce the problem. We have re-created the file and believe the issue has been resolved.

Lines 199-206

Authors describe alkylating agents, but they only mention mustard. Indeed, LAT1 also interacts with other alkylating agents. Therefore, unless authors have more than one example, PGA should be treated as a single case study or other data from literature should be cited.

Answer:

Thank you for pointing out the existence of various alkylating agents beyond mustard. We have revised the last paragraph of this section, which now mentions a broader range of alkylating reagents that interact with LAT1 and introduces mustard as one example.

Line 208.

The role of BCH as a substrate and, therefore, as a competitive inhibitor (see line 69 of introduction) is not an original data (see point above). This needs to be well clarified also in the view of the relationships with the structure. Fig. 4 h should be modified; since BCH is a substrate it induces the normal antiport function not a trans-stimulation phenomenon.

Answer:

We acknowledge that our original manuscript did not adequately mention the known role of BCH as a substrate. We have revised the paragraph to explicitly state this fact and cite previous experiments. For figures, we have changed "trans stimulation" to "induce antiport", and "inhibitor" to "test compound" to reflect these changes. We would like to note that the term "trans stimulation" is commonly used in the transporter literature to describe the phenomenon whereby substrate transport is stimulated by the presence of a certain compound on the other side of the membrane [Geier et al. PNAS, ref 21]. However, we recognize that this terminology is cell biology oriented and may not be appropriate for structural biology/biochemistry, where "trans" could be interpreted as an allosteric effect. Therefore, we have removed this term throughout the manuscript and instead use "induce the antiport" or similar.

The manuscript will benefit from a good effort in discussing the basis for transport of this compound in relationships to the occluded conformation (they only try on line 231). Is it a fully occluded state, or is it occluded but oriented toward a side of the membrane etc? An appropriate interpretation/discussion of data would be of great interest for readers.

Answer:

The occluded conformation bound to BCH is slightly outward-facing, as judged by the slight opening of TM1b and TM6a compared to those in the inward-facing conformation. However, BCH is not exposed to the extracellular solvent, in contrast to the previously reported outward-occluded state bound to JX inhibitors [Yan et al., Cell Discovery, 2020]. Therefore, we believe that the simple "occluded" state is an appropriate description for our BCH-bound structure. At this point we do not know if a hypothetical "fully occluded" conformation other than the one observed here would exist, since the substrate pocket is already just tight enough for BCH, and further closure of TM1b and TM6a would be sterically unfavorable.

Lines 234-236.

Again, this part should be rewritten on the basis of the finding that BCH is a substrate.

Answer:

We have revised this part to first introduce the previous work by Geier et al. (PNAS, 2013) and rephrased the word "demonstrate" to "confirm". Perhaps the difference here is that we measured the amount of external substrates to confirm efflux, whereas the previous work only measured intracellular substrates, and that we used *Xenopus* oocytes, which have lower background transport activity, allowing us to achieve a higher signal-to-background ratio.

Line 243.

This is true, but not an original result, please cite the appropriate previous findings (see above).

Answer:

We have corrected the text to use the words like "confirm" instead of "reveal".

Lines 248-253.

Is it possible that this state represents the more stable conformation, a sort of ground state? Authors completely ignore previous findings on the functional asymmetry of LAT1. Previous literature and specific reviews on LAT1 on this topic could be very useful for a more interesting and appropriate discussion of the reported 3D data.

Answer:

This is an interesting point. Yes, we think this may be the ground state, since we and others have only observed the inward-facing state in detergent and in the absence of ligands. We have added a paragraph discussing the functional asymmetry, citing the previous findings on the functional asymmetry of LAT1 in the Discussion.

Lines 278-283.

The finding that the protein lacking 1-50 residues is not active suggests that this portion of LAT1 is relevant to the transport mechanism or its regulation. Are there PTM sites? This issue should be better discussed and clarified.

Answer:

Thank you for the idea of post-translational modification and the suggestion to clarify how residues 1-50 might be involved in the transport mechanism. Indeed, previous studies have shown that three lysine residues in the N-terminal region (K19, K25, and K30) are critical for down-regulation of LAT1 by ubiquitination [doi: 10.1038/s41598-019-53065-w]. However, there is no evidence that they are involved in the transport mechanism itself. We speculate that since residues 44-50 make critical cytoplasmic interactions in the outward-facing and occluded conformations, the deletion resulted in the arrest of the transporter in the inward-facing conformation and thus no transport function.

Lines 291-292.

Please clarify this part also in view of previous literature data on cholesterol interactions.

Answer:

We have added references suggested in the previous comments that have shown important biochemical roles of cholesterol. Notably, the putative binding site predicted in that study (based on the homology model from the *Drosophila* dopamine transporter) is completely different from the one observed here. We have added a remark to this fact. We still believe that previous biochemical data are valid because they are independent of the binding site prediction. Notably, a recent docking study published during the revision [<https://doi.org/10.1021/acs.jctc.3c01391>] showed a cholesterol stably bound at the exact same position as CHOL3 in our structure. This is now included in the Discussion, along with another mass spectrometry study.

Discussion

The discussion section is poor being a list of the main findings. Indeed, this section does not add any new information or speculation/hypotheses about the results. It should be rewritten also considering the observations on Results.

Answer:

We acknowledge that we limited the Discussion section of the original manuscript to a summary of the main findings and did not discuss speculation or new hypotheses. Based on the suggestion, we have added some ideas and speculative thoughts that we believe are of broad interest to the field.

Specific comments:

Lines 299-300.

Please discuss the similarities/advancements with respect to the previously 3D structures and previous published computational predictions of LAT1 and interactions with substrates and inhibitors.

Answer:

Thank you very much for the suggestion to improve the discussion section. The revised manuscript now discusses similarities, differences and advances that our study provides with respect to the previous experimental structures and computational predictions.

Line 303.

If BCH is a substrate, as it is, it does not affect the transporter turnover (it is not an allosteric activator). It just induces the antiport function.

Answer:

We acknowledge this point in accordance with the related comments above. We corrected this part as well as all similar expressions.

"... that can stimulate transporter turnover."

→ "... that can induce the antiport function."

Response letter to the reviewers

Reviewer #1 (Remarks to the Author):

The study by Lee et al. report the structure and characterization of LAT1 inhibition by several compounds. The authors determine structures of the protein in multiple states and compound-bound complexes. The topic itself is important and significant experiments appear well-designed, and extensive data was collected to address their hypotheses.

The authors have done a commendable job resolving my technical and scientific concerns I had in the previous review. I have no more concerns and am happy for this paper to proceed to publication.

Answer:

We appreciate again the careful review and valuable comments, which helped improve the manuscript.

Reviewer #3 (Remarks to the Author):

The responses of the authors to my concerns were appropriate and the manuscript has been changed accordingly. Only one issue should be dealt with more extensively, that is the double mutant issue (now lines 151-155). The results should be better explained. Some comments, based on the author hypothesis, should be added in the discussion or in the results section. After this change the manuscript can be acceptable.

Answer:

Thank you for carefully reviewing the revised manuscript and raising a remaining issue. To clarify the double mutant issue, we have added a sentence elaborating on our interpretation of the results in the corresponding paragraph, as quoted below. We have also slightly modified the preceding sentence to explain why we could interpret Ser144 as the key residue for the JPH203 selectivity from the results of the single and double mutants.

Line 154–160:

“Although a single variant S144N abolished L-Leu transport, and thus its own effect on inhibition could not be evaluated independently (Fig. S4a), the variant F400V retained its activity and was sensitive to JPH203 (Fig. S4a,b), leaving Ser144 as a key residue for JPH203 selectivity. We reason that the two mutations of opposite effects (smaller to larger and larger to smaller) compensated for each other to restore the binding pocket for amino acid substrates (as seen in the LAT2 structures (24, 25)), whereas this compensation was not sufficient for JPH203 due to stricter recognition.”